# Mid-Term Results of Ab Interno Trabeculectomy among Japanese Glaucoma Patients

**DOI:** 10.3390/jcm12062332

**Published:** 2023-03-16

**Authors:** Kazuyoshi Kitamura, Yoshiko Fukuda, Yuka Hasebe, Mio Matsubara, Kenji Kashiwagi

**Affiliations:** Department of Ophthalmology, Faculty of Medicine, University of Yamanashi, Chuo 409-3898, Japan

**Keywords:** ab interno trabeculectomy, glaucoma, intraocular pressure

## Abstract

Background: The evaluation of ab interno trabeculectomy, referred to as trabectome^®^, among Japanese patients is insufficient. Subjects and methods: Japanese patients who underwent trabectome^®^ at the University of Yamanashi Hospital were included. The investigated parameters were intraocular pressure (IOP), best corrected visual acuity, glaucoma medications, visual field, and corneal endothelial cell density. The success rate and its associated factors were investigated. Results: A total of 250 eyes from 197 patients were enrolled. The trabectome^®^ significantly reduced IOP and glaucoma medications up to 48 months. Concomitant cataract extraction enhanced the reduction in IOP and glaucoma medications up to 42 months. At 36 months postoperatively, 40.8% satisfied IOP of the same or less than 18 mmHg or more than a 20% IOP reduction with the same or less use of glaucoma medications as preoperatively. Preoperative IOP and combined cataract extraction were significantly associated with the success rate. The trabectome^®^ alone did not show a significant reduction in corneal endothelial cells. Eyes with postoperative transient IOP elevation and removal of anterior chamber hemorrhage were 11.2% and 1.2%, respectively. Twenty-four eyes (9.6%) underwent additional glaucoma surgeries. Conclusions: The trabectome^®^ could be considered an effective and safe surgery. Compared to trabectome^®^ alone, combined cataract surgery was superior in lowering IOP and reducing glaucoma medications.

## 1. Introduction

Glaucoma is a high-ranking cause of blindness with irreversible optic neuropathy. Although glaucoma eye drops are commonly used to lower intraocular pressure (IOP), surgical treatment is often required due to inadequate IOP reduction or side effects. Trabeculectomy is a major glaucoma surgery. Though it is beneficial for lowering IOP, it is a procedure with a high risk of postoperative complications such as bleb infection, complications related to low IOP, and expulsive bleeding [1]. Trabectome^®^ is an ab interno trabeculectomy and is considered a minimally invasive glaucoma surgery (referred to as MIGS) that is performed through a small corneal incision. Trabectome^®^ was approved by the FDA in 2004 and by the Japanese Ministry of Health, Labor, and Welfare in 2010.

The purpose of this study was to evaluate the mid-term efficacy and safety of trabectome^®^.

## 2. Materials and Methods

### 2.1. Study Design

All patients who underwent trabectome^®^ at the University of Yamanashi Hospital between March 2016 and December 2017 and met the following criteria were included in the retrospective study. The study was approved by the Ethics Committee of the University of Yamanashi School of Medicine. It was performed in accordance with the Declaration of Helsinki:

Approval Code: 1974;

Approval Date: 20 March 2019.

### 2.2. Entry and Exclusion Criteria

Entry criteria included consecutive patients aged 20 years or older who underwent trabectome^®^ at the University of Yamanashi Hospital between March 2016 and December 2017. Accurate intraocular pressure measurement by Goldman Applanation Tonometry (GAT) was possible before and after surgery. Patients who underwent surgery on both eyes were included in the analysis.

### 2.3. Surgical Technique and Postoperative Management

Surgery was performed by three surgeons (K.K., M.M., and K.K.) using the trabectome^®^ system (Neomedix, Inc., Tustin, CA, USA) at the University of Yamanashi Hospital. In the case of pseudophakic eyes before surgery, Trabectome^®^ alone is indicated. In the case of phakic eyes before surgery, combined cataract surgery with trabectome^®^ is indicated when cataracts are recognized. The presence or absence of cataracts was determined by the surgeon before surgery. A 1.7 mm temporal corneal incision was made, viscoelastic material was inserted intraocularly, the trabectome^®^ was inserted, and a Swan Jacob Gonioprism was used to resect the nasal trabecular meshwork in the 90–120-degree range. In the trabectome^®^ alone group, the surgery was completed after aspiration and removal of reflux hemorrhage and residual viscoelastic material using the trabectome^®^. In the combined cataract surgery group, conventional lens reconstruction was performed through the same corneal incision. After surgery, steroid eye drops (Betamethasone Sodium Phosphate Ophthalmic Solution 0.1%) and topical antibiotic eye drops (Levofloxacin Ophthalmic Solution 1.5%) were administered four times a day for one month, and 2% pilocarpine hydrochloride ophthalmic solution was started four times a day and gradually decreased. In the combined cataract group, NSAID eye drops (Bromfenac Sodium Hydrate Ophthalmic Solution 0.1%) were used twice a day for 3 months after surgery. All preoperative glaucoma eye drops and oral medications were discontinued, and additional medications were added at the discretion of the surgeon when an increase in IOP was observed.

### 2.4. Outcome Measures and Safety Evaluation

The primary outcome measures in this study were IOP values measured by GAT and the number of glaucoma medications. Preoperative data were collected based on the last preoperative visit. Patients used glaucoma medications until the day before surgery, with no washout period. The number of glaucoma medications was counted as two for combination eye drops and one for oral carbonic anhydrase inhibitors. Postoperative data were collected from patients at 1 week, 2 weeks, 1 month, and every 3 months thereafter. Secondary endpoints included visual acuity tests, visual field tests, and corneal endothelial cell density before and after surgery. Safety endpoints included surgical complications, adverse events, and additional glaucoma surgeries. Visual field measurements were performed using a Humphrey visual field analyzer (HFA) SITA-standard (Carl Zeiss Meditec, Dublin, CA, USA), and mean deviation (MD) values were included in the analysis.

### 2.5. Statistical Analyses

Statistical analyses were performed using JMP12.0.1 (SAS Institute, Cary, NC, USA), and changes in IOP and glaucoma medications before and after surgery were analyzed using repeated ANOVA. Corneal endothelial cell density before and after surgery was analyzed using a paired *t*-test. Changes in visual field defects before and after the surgery were compared using a paired *t*-test or repeated ANOVA. Factors related to survival were analyzed using logistic regression analysis. A *p* value < 0.05 was considered statistically significant.

## 3. Results

### 3.1. Demographics and Preoperative Ocular Parameters

Table 1 shows the preoperative data of the patients studied. The study population consisted of 107 male and 90 female patients with a mean age of 70.7 ± 12.6 years (16–93 years). The postoperative observation period ranged from 1 to 60 months. There were 201 phakic eyes and 48 pseudophakic eyes. There were 154 eyes that underwent combined cataract surgery with trabectome^®^ and 96 eyes that underwent trabectome^®^ alone. The preoperative IOP was 20.6 ± 6.8 mmHg.

Glaucoma types were primary open-angle glaucoma (POAG) in 158 eyes (63%), pseudoexfoliative glaucoma (PXG) in 41 eyes (16%), neovascular glaucoma (NVG) in 2 eyes (1%), mixed glaucoma in 8 eyes (3%), secondary glaucoma (SG) in 31 eyes (12%), and glaucoma of undetermined type in 10 eyes (4%). We performed trabectome^®^ for open-angle glaucoma without extensive peripheral anterior synechia (PAS). In the case of NVG, Trabectome is performed if there is no active angle neovascularization. Patients with NVG showed burn-out status of their NGV, and we confirmed that their angle was open without severe PAS formation. We used topical oxybuprocaine hydrochloride eyedrops for anaesthetizing. The mean preoperative best corrected visual acuity (BCVA) (logMAR) was 0.20 ± 0.36, and the mean preoperative HFA24-2MD value was −10.5 ± 7.0 dB (1.31 to –31.48). The mean number of glaucoma medications administered preoperatively was 4.1 ± 1.3.

### 3.2. Changes in IOP and Medications

Figure 1 shows the mean IOP before and after surgery (Figure 1a) and the rate of IOP reduction (Figure 1b). The mean IOP decreased from 20.6 ± 6.8 mmHg preoperatively to 15.0 ± 4.1 mmHg at 6 months, 15.6 ± 4.0 mmHg at 12 months, 15.6 ± 3.2 mmHg at 24 months, 16.5 ± 5.3 mmHg at 36 months, and 17.5 ± 6.2 mmHg at 48 months after surgery, and the differences in decreases were statistically significant from 6 to 51 months postoperatively (*p* < 0.05). The IOP reduction rate averaged 23.4% for the entire postoperative period, ranging from 20 to 25%.

Figure 2 shows the number of glaucoma medications (Figure 2a) and the percentage of patients who were medication free (Figure 2b) before and after surgery. The number of glaucoma medications decreased from 4.1 ± 1.3 preoperatively to 1.5 ± 1.4 at 6 months, 1.5 ± 1.6 at 12 months, 1.9 ± 1.6 at 24 months, 2.4 ± 1.8 at 36 months, and 2.7 ± 1.9 at 48 months after surgery. The mean of all postoperative periods was significantly lower (*p* < 0.001). The average decrease in the number of glaucoma medications during the entire postoperative period was 48.0% (−1.96). The number of postoperative glaucoma medications tended to gradually increase over time (Figure 2a). The percentage of postoperative glaucoma medication-free patients was up to 38.3% at 9 months and 38.2% at 12 months, followed by a gradual decrease (Figure 2b).

### 3.3. Preoperative IOP and IOP Reduction

Figure 3 shows the results of the postoperative IOP changes after dividing the preoperative IOP into three groups: Group A: less than 15 mmHg, Group B: 15–20 mmHg, and Group C: 21 mmHg or higher. Pre-IOP value means last preoperative visit IOP. Postoperative IOP in Groups B and C significantly decreased for most of the time periods (Group B: 1D-54 M (*p* < 0.05), Group C: 1D-57 M (*p* < 0.05)), whereas Group A showed no significant decrease in all measured time periods. The number of postoperative medications decreased significantly in all three groups for most of the time periods compared to the preoperative period (Group A: 1D-39 M, 48 M (*p* < 0.05); Group B: 1D-45 M (*p* < 0.05); Group C: 1D-54 M (*p* < 0.05)).

### 3.4. Success Rate

Figure 4 shows the results of the success rate study as defined below.

Complete Success was defined as no glaucoma medications, IOP less than 15 mmHg, and at least a 20% reduction from preoperative IOP.

Qualified success 1 was defined as IOP less than or equal to 15 mmHg and at least a 20% reduction from preoperative IOP. Postoperative glaucoma medications were equal to or less than preoperative.

Qualified success 2 was defined as IOP less than or equal to 18 mmHg and at least a 20% reduction from preoperative IOP. Postoperative glaucoma medications were equal to or less than preoperative.

The percentage of patients who achieved complete success was 25.4% at 12 months, 13.1% at 24 months, 6.9% at 36 months, 4.6% at 48 months, and 4.6% at 60 months.

The percentage of patients who achieved qualified success 1 was 39.2% at 12 months, 27.6% at 24 months, 24.6% at 36 months, 16.1% at 48 months, and 10.6% at 60 months.

The percentage of patients who achieved qualified success 2 was 60.6% at 12 months, 49.6% at 24 months, 42.5% at 36 months, 36.0% at 48 months, and 32.2% at 60 months.

The percentage of patients who achieved qualified success 2 in the POAG and PXG groups and for other types of glaucoma was evaluated (Figure 5a).

In the POAG and PXG groups, 74.9% of patients achieved qualified success 2: 61.1% at 12 months, 49.6% at 24 months, 40.3% at 36 months, 32.9% at 48 months, and 29.3% at 57 months.

In the other glaucoma groups, the rates of qualified success 2 were 58.2% at 12 months, 49.5% at 24 months, 43.1% at 36 months, 39.8% at 48 months, and 39.8% at 57 months.

Comparison using the log-rank test between the POAG and PE groups and the other glaucoma groups showed no significant difference in survival rates. There were also no significant differences in survival rates between the POAG and PXG groups (*p* = 0.35).

Next, the percentage of patients who achieved qualified success 2 in the combined cataract group and the Trabectome^®^ alone group was examined (Figure 5b).

In the combined cataract group, the percentage of patients who achieved qualified success 2 was 69.0% at 12 months, 61.0% at 24 months, 51.1% at 36 months, 47.2% at 48 months, and 43.3% at 60 months.

In the Trabectome^®^ alone group, the percentage of patients who achieved qualified success 2 was 46.9% at 12 months, 31.8% at 24 months, 24.7% at 36 months, 15.2% at 48 months, and 39.8% at 54 months.

When compared using the log-rank test for the combined cataract and trabectome^®^ alone groups, the combined cataract group had a significantly higher survival rate (*p* < 0.0001).

### 3.5. Examination of Factors Related to Success

Factors associated with success in the above definition were examined. Logistic regression analysis was used to examine factors associated with success at 3 years postoperatively, including age, sex, glaucoma type, trabectome^®^ alone/combined cataract, preoperative IOP, number of preoperative medications, and postoperative complications.

There were no significant factors associated with complete success, including age (*p* = 0.19), sex (*p* = 0.09), glaucoma type (*p* = 0.93), trabectome^®^ alone/combined cataract (*p* = 0.41), preoperative IOP (*p* = 0.36), preoperative medications (*p* = 0.45), or postoperative complications (*p* = 0.96).

Preoperative IOP was significantly associated with qualified success 1 (*p* = 0.02). No significant associations were found for other factors, including age (*p* = 0.65), sex (*p* = 0.47), glaucoma type (*p* = 0.64), trabectome^®^ alone/combined cataract (*p* = 0.41), number of preoperative medications (*p* = 0.72), or postoperative complications (*p* = 0.62).

The presence of combined cataract surgery was significantly associated with qualified success 2 (*p* < 0.01). No significant associations were found for other factors, including age (*p* = 0.80), sex (*p* = 0.98), glaucoma type (*p* = 0.90), preoperative IOP (*p* = 0.40), number of preoperative medications (*p* = 0.70), or postoperative complications (*p* = 0.44).

### 3.6. Comparison between Combined Cataract Surgery and Trabectome^®^ Alone Surgery

Figure 6 shows the changes in IOP (Figure 6a) and the number of glaucoma medications (Figure 6b) in the combined cataract group and the trabectome^®^ alone group. In the combined cataract group, IOP significantly decreased from the day after surgery to 48 months postoperatively, and in the trabectome^®^ alone group, IOP significantly decreased from the day after surgery to 45 months postoperatively (*p* < 0.05). The mean postoperative IOP reduction rate was 22.2% in the combined cataract group and 24.0% in the trabectome^®^ alone group, with both groups showing greater IOP reductions, significantly more so in the combined cataract group (*p* < 0.001).

The number of glaucoma medications also significantly decreased from 1 week to 51 months postoperatively in the combined cataract group and significantly decreased from 1 week to 39 months postoperatively in the trabectome^®^ alone group (*p* < 0.05). The mean postoperative reduction in the number of drugs was 55.4% (−2.0) in the combined cataract group and 37.9% (−1.8) in the trabectome^®^ alone group, which was significantly higher in the combined cataract group (Figure 6b).

The mean postoperative reduction in IOP was similar in both groups, but the decrease in the number of glaucoma medications was greater in the combined cataract group. A comparison of the two groups at each measurement period showed that IOP was significantly lower in the combined cataract group than in the trabectome^®^ alone group for most periods from preoperative to 42 months postoperative (*p* < 0.05). In addition, the number of glaucoma medications was significantly lower in the combined cataract group at all time points from preoperative to 42 months (*p* < 0.01).

### 3.7. Changes in BCVA and Visual Field

Figure 7 shows the changes in BCVA in the combined cataract surgery group and the trabectome^®^ alone group. BCVA significantly improved in the combined cataract group from 1 to 30 months postoperatively (*p* < 0.05). The BCVA significantly decreased in the trabectome^®^ alone group (*p* < 0.001) during the first postoperative week, but the significant difference from the preoperative period disappeared from 2 weeks to 57 months postoperatively (Figure 7).

Overall, HFA24-2MD values were −10.5 ± 7.0 dB (1.31–31.48) preoperatively and −10.3 ± 7.0 dB (1.69–28.46) at 12 months postoperatively, with no significant deterioration in visual field defects (*p* = 0.81, paired *t*-test). In the long-term follow-up, HFA24-2MD values were 10.2 ± 7.1 dB (0.96–27.61) at 24 months postoperatively, 11.4 ± 7.4 dB (0.08–29.93) at 36 months postoperatively, and 11.4 ± 7.8 dB (−0.67–30.77) at 48 months postoperatively, with no significant deterioration of any visual field defect (repeated ANOVA).

The HFA24-2MD value was −11.7 ± 6.5 dB (−0.02–31.48) preoperatively and −11.4 ± 6.9 dB (−1.12–28.46) at 12 months postoperatively in the combined cataract group, with no significant deterioration of visual field defects (*p* = 0.72).

In the trabectome^®^ alone group, the preoperative HFA24-2MD value was −8.77 ± 7.2 dB (1.31–29.79), and the 12-month postoperative value was −8.84 ± 7.0 dB (1.69–27.33), showing no significant deterioration of visual field defects (*p* = 0.95).

### 3.8. Safety Profile

Surgical complications are shown in Table 2. Postoperative transient elevation of IOP (≥30 mmHg) was observed in 28 eyes (11.2%). The transient elevation of IOP occurred from the day after surgery to 1 month after surgery. Transient elevation of IOP occurred in 4 (2.6%) eyes in the combined cataract group and 24 (25%) eyes in the trabectome^®^ alone group, with a significantly higher incidence in the trabectome^®^ alone group (*p* < 0.01). Severe anterior chamber hemorrhage requiring anterior chamber wash out was observed in three eyes, all in the trabectome^®^ alone group. No serious complications, such as endophthalmitis, expulsive bleeding, or choroidal detachment, were observed.

Of the 24 eyes that underwent additional glaucoma surgery due to poor postoperative IOP control, most (19) underwent trabeculectomy, 3 underwent additional trabectome^®^, 1 underwent trabectome^®^ + goniosynechialysis (GSL), and 1 underwent trabeculotomy. The average time of additional surgery was 27.8 months postoperatively (2 weeks to 42 months postoperatively). We excluded the following information from analysis at the time of additional surgery. Five eyes (2%) underwent additional procedures: yttrium aluminum garnet (YAG) in one eye, cyclo photo coagulation (CPC) in two eyes, and selective laser trabeculoplasty (SLT) in two eyes, ranging from 1 month to 21 months postoperatively.

Changes in corneal endothelial cell density are shown in Table 3. In all patients, corneal endothelial cell density significantly decreased from 2465.6 ± 376.3/mm^2^ preoperatively to 2378.6 ± 378.2/mm^2^ postoperatively (−3.52%) (*p* < 0.001). The combined cataract group significantly decreased from 2518.4 ± 333.6/mm^2^ preoperatively to 2381.0 ± 359.9/mm^2^ postoperatively (−5.45%) (*p* < 0.001), whereas the trabectome^®^ alone group showed no significant difference, from 2353.4 ± 435.5/mm^2^ preoperatively to 2374.6 ± 408.2/mm^2^ postoperatively. Therefore, the decrease in corneal endothelial cell density was considered to be an effect of cataract surgery.

## 4. Discussion

### 4.1. Summary of Results: Intraocular Pressure and Number of Medications

This study examined the postoperative results of trabectome^®^ surgery in Japanese glaucoma patients treated at our hospital up to 60 months postoperatively. IOP and the number of glaucoma medications were significantly decreased during most of the follow-up period compared to the preoperative period. It is thought that the reason for which the significant difference disappeared after 50 months was due to the decrease in the number of subjects for analysis. The average IOP reduction was 23.4% for the entire period, ranging from 20 to 25%, and the reduction in the number of glaucoma medications during the entire postoperative period was 48.0%. The percentage of patients who were free of glaucoma medication after surgery was 38.2% at 12 months, which continued to gradually decrease. IOP reduction was greater in the combined cataract surgery group, and the number of glaucoma drugs decreased substantially.

In previous reports by Avar M, Minckler D et al., a baseline IOP of approximately 23.0 mmHg was reduced to 16.5–17.2 mmHg (−26 to −28%) [2,3]. Esfandiari et al. and Bendel et al. reported a decrease in baseline IOP from 18.0~20.0 mmHg to 13.9~15.6 mmHg (−19~22%) [4,5]. In a study of Japanese subjects, Kono et al. reported a reduction from a baseline IOP of 29.2 mmHg to 16.4 mmHg (−43.8%) after 72 months [6]. Tojo et al. also reported a decrease from a baseline IOP of 23.0 mmHg to 13.6 mmHg after 24 months [7]. In our results, the rate of IOP reduction was somewhat less than that previously reported, but this may be because the preoperative IOP in our study was somewhat lower than that in other reports. The postoperative IOP was almost the same as that in a previous report. In both our study and previous reports, the rate of IOP reduction was lower than that of trabeculectomy [8]. Trabectome^®^ is a surgical technique that promotes aqueous humor outflow in the main aqueous humor pathway by resecting the trabecular meshwork, and the outflow resistance of the outflow tracts (collecting channel and superior scleral vein) after the trabecular meshwork is constant, which may explain why the lower limit of IOP reduction was higher than that in trabeculectomy.

In the present study, preoperative IOP was divided into three groups: Group A: <15 mmHg, Group B: 15–20 mmHg, and Group C: >20 mmHg, and postoperative IOP changes were examined. The number of glaucoma medications in all three groups decreased significantly, and IOP also decreased significantly in groups A and B. However, in group C, there was no significant decrease in all postoperative periods. These results are similar to those of Tojo et al. and suggest that trabectome^®^ should be indicated in patients with a preoperative IOP of 16 mmHg or higher [7].

Previous reports have reported higher postoperative success rates in the combined cataract surgery group than in the trabectome^®^ alone group, similar to our results [6,9,10]. This may be because cataract surgery itself has an IOP-lowering effect, and the increase in IOP during cataract surgery after trabectome^®^ surgery reduces postoperative anterior chamber hemorrhage and transient IOP elevation [11].

### 4.2. Factors Involved in Trabectome^®^ Outcome

In this study, IOP control was considered to be successful at 3 years postoperatively in 6.9% of patients in the complete success group, 20.6% in qualified success group 1, and 40.8% in qualified success group 2. An analysis of the factors associated with success showed no significant association in the complete success group, but preoperative IOP was significantly associated with success in qualified success group 1 (*p* = 0.02), and the presence or absence of combined cataract surgery in qualified success group 2 was significantly associated (*p* < 0.01). Preoperative IOP may affect the success rate of the trabectome^®^, and therefore, patients with high preoperative IOPs should be treated with the possibility of poor success rates in trabectome^®^ procedures. In addition, the presence or absence of combined cataract surgery is significantly related to the success rate of trabectome^®^ and should be actively considered in patients who are eligible for combined cataract surgery.

Kono et al. reported that 26% of their patients had an IOP of 16 mmHg and achieved an IOP reduction of 20% or greater at 3 years postoperatively, and 46% had an IOP of 18 mmHg and achieved an IOP reduction of 20% or greater, with our results showing a slightly lower success rate [6]^6^. In addition, our results showed no significant differences by glaucoma type, and the survival rate was significantly higher in the combined cataract group than in the trabectome^®^ alone group, all of which are similar to previously reported results.

### 4.3. Advantages of the Trabectome^®^

Among the subjects in the current study, up to 38.3% of the patients did not require postoperative glaucoma medications. Although glaucoma medication has made significant progress in recent years, problems associated with multiple-medication therapy, such as poor adherence, dropout from medication therapy, and side effects of eye drops and oral medications, have become apparent. Therefore, very few medications should be used. In the present study, a maximum of 38.3% of patients achieved medication-free treatment, and the number of glaucoma medications was reduced from the preoperative level in most patients. This is very useful for glaucoma treatment in terms of improving adherence, preventing dropouts, and reducing the side effects of eye drops and oral medications.

### 4.4. Summary of Complications and Comparison with Previous Reports

In the present study, transient postoperative IOP elevation (≥30 mmHg) was observed in 28 eyes (11.2%), and severe anterior chamber hemorrhage requiring anterior chamber wash out was observed in 3 eyes, but there were no cases of serious complications, such as endophthalmitis or expulsive bleeding, which were similar to those in previous reports [12,13]. Compared to the combined cataract group, the rate of transient postoperative IOP elevation and severe anterior chamber hemorrhage was higher in the trabectome^®^ alone group, but this may be because, as mentioned above, the increase in IOP during cataract surgery can reduce transient IOP elevation by reducing postoperative anterior chamber hemorrhage. Trabectome^®^ is a safe, low-patient-load technique with low surgical invasiveness and an infrequent need for postoperative surgical procedures (9.6% of patients required such procedures).

In addition, corneal endothelial cell density was significantly decreased in all patients and in the combined cataract group but not in the trabectome^®^ alone group, suggesting that the effects of cataract surgery were not significant, and that trabectome^®^ caused little damage to corneal endothelial cells.

### 4.5. Limitations of the Present Study

Limitations of the current study include the following: the lack of a control group in a retrospective study. There were three surgeons, and there may be differences in surgical techniques among the surgeons. Although the results were obtained during a period of up to 60 months, the follow-up period varied from patient to patient. Multiple types of disease were entered. There are no criteria for the resumption of glaucoma medication after surgery, and resumption is at the discretion of the surgeon. The corneal endothelial cell density was not compared to that in the cataract surgery group.

However, trabectome^®^ is a safe procedure with a short operative time and few intraoperative and postoperative complications.

Trabectome^®^ is suitable for patients who are unable to continue long-term medication therapy, who are concerned about side effects or poor adherence to multiple medications, or who are expected to drop out of treatment. However, since this is a surgical procedure to promote aqueous humor outflow in the main aqueous humor pathway, and since the outflow resistance of the outflow tracts (collecting channels and superior scleral vein) after the trabecular meshwork is constant, there is a lower limit to IOP reduction and a limit to IOP reduction value, so care should be taken in selecting cases. Trabeculectomy should be carefully considered for patients whose IOP before surgery is stable at 15 mmHg or lower but whose visual field is progressing or whose glaucoma is at the end stage and whose target IOP of approximately 10 mmHg is desired.

In addition, our results showed that trabectome^®^ was performed in patients who were using more than four medications before surgery and in patients who were using multiple types of medications. As the number of glaucoma medications increases, patient adherence declines, and the side effects of eye drops worsen. In addition, the results of the present study showed that the combined cataract group had greater IOP reductions and fewer drugs than the trabectome^®^ alone group. Therefore, the use of trabectome^®^ in patients who require cataract surgery may be actively considered regardless of glaucoma type. In the future, we may consider performing trabectome^®^ surgery for a wide range of indications, from patients with early-stage glaucoma who are receiving single-drug therapy to patients with end-stage glaucoma who cannot undergo trabeculectomy due to advanced age or poor general condition. Kashiwagi et al. reported that trabectome^®^ surgery, both in combined cataract surgery and as a stand-alone procedure, not only lowers IOP but also improves practical vision by improving ocular surface conditions such as corneal epithelial damage due to a decrease in the number of medications [14]. We believe that this may further contribute to the improvement of patient QOV after surgery.

In Japan, following trabectome^®^, various MIGS, such as iStent^®^, iStent inject W^®^, Kahook dual blade^®^, and μ-hook^®^, are now available.

According to Iwasaki et al., IOP significantly decreased from 19.8 ± 7.3 mm Hg to 13.0 ± 3.1 mm Hg, and the mean number of medications significantly decreased from 2.5 ± 1.4 to 1.6 ± 1.6 in the Kahook Dual Blade^®^ procedure. IOP significantly decreased from 17.8 ± 2.9 mmHg to 14.3 ± 2.3 mmHg, and the mean number of medications significantly decreased from 2.2 ± 1.1 to 0.9 ± 1.4. The IOP reduction rate was 26.2% with Kahook Dual Blade^®^ and 19.0% with iStent^®^ surgery [15], which were comparable to our results.

Nitta et al. reported that the iStent procedure with cataracts reduced IOP by 18% from 16.5 ± 3.4 mmHg preoperatively to 13.6 ± 3.0 mmHg and reduced the number of glaucoma drugs by 81% from 1.96 ± 0.98 preoperatively to 0.37 ± 0.74, with 77% of the patients being glaucoma medication free [16]. Although our results showed a greater IOP reduction, the percentage of patients who became glaucoma medication free was lower. The reason for this may be that the number of preoperative glaucoma medications in our study was higher, and although the number of medications could be significantly reduced, the patients did not become glaucoma medication free.

iStent inject W^®^ has been available in Japan since 2020. Overseas reports have shown that iStent inject W^®^ is superior to iStent^®^ in lowering IOP and significantly reducing the number of glaucoma medications [17,18]. There are also reports that iStent inject W^®^ and trabectome^®^ are equivalent and others that iStent inject W^®^ is superior to trabectome^®^ [19,20]. In Japan, there are still few reports on iStent injection W^®^, so further studies are needed, including comparisons with trabectome^®^.

Tanito et al. reported that IOP decreased from 25.9 mmHg preoperatively to 14.5 mmHg at 6 months postoperatively for μ-hook surgery alone and from 16.4 ± 2.9 mmHg preoperatively to 11.8 ± 4.5 mmHg at 9.5 months postoperatively for combined cataract surgery [21,22]. In addition, Tojo et al. compared surgical outcomes between μ-hook and trabectome^®^ and reported that trabectome^®^ had significantly better surgical outcomes, although no significant difference was found for postoperative IOP [23].

Although there are differences in the results of IOP reduction and reduction in the number of glaucoma medications among the various techniques, we believe that the various MIGS techniques are minimally invasive and useful, resulting in good reductions in IOP and the number of glaucoma medications and fewer surgical complications.

## 5. Conclusions

Trabectome^®^ is a minimally invasive and useful procedure that leads to a reduction in the number of glaucoma medications without serious complications, a significantly lower IOP, and improved visual acuity compared to preoperative visual acuity. Compared to trabectome^®^ surgery alone, combined cataract surgery was superior in lowering IOP and reducing the number of glaucoma medications. However, there is a limit to the amount of IOP reduction, so the indication for trabectome^®^ surgery should be carefully evaluated in patients with low target IOP.

## Figures and Tables

**Figure 1 jcm-12-02332-f001:**
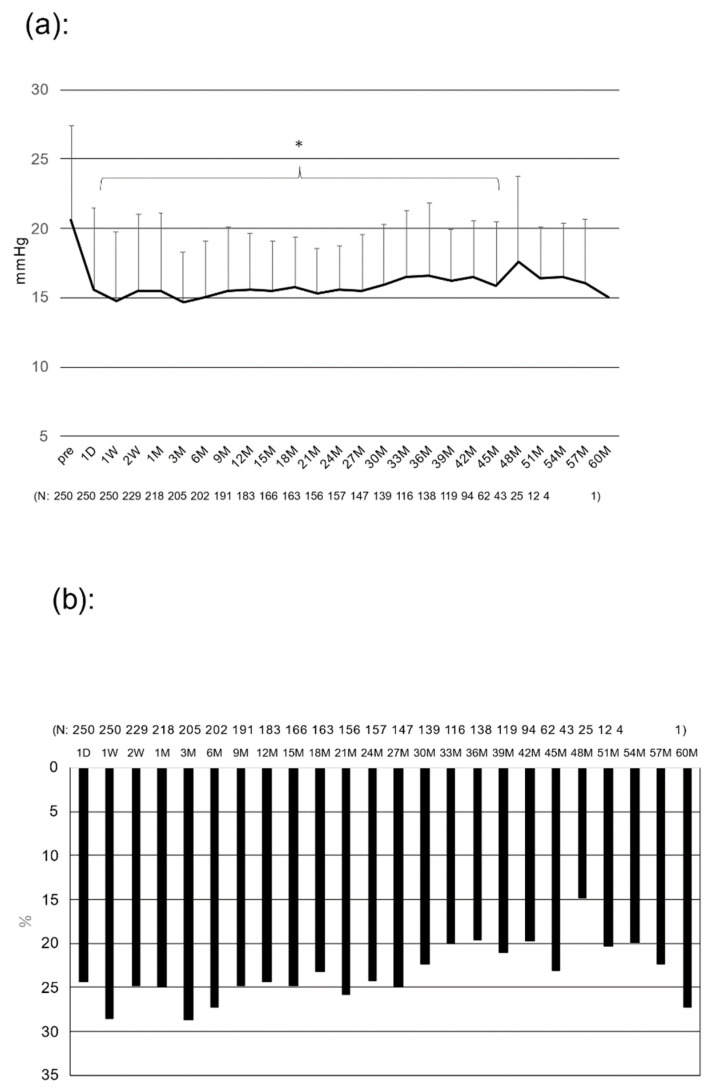
Postoperative changes in IOP and IOP reduction rate: (**a**) postoperative changes in IOP, * *p* < 0.05 vs. preoperative value (repeated ANOVA), (**b**) postoperative IOP reduction rate. Bar = SD.

**Figure 2 jcm-12-02332-f002:**
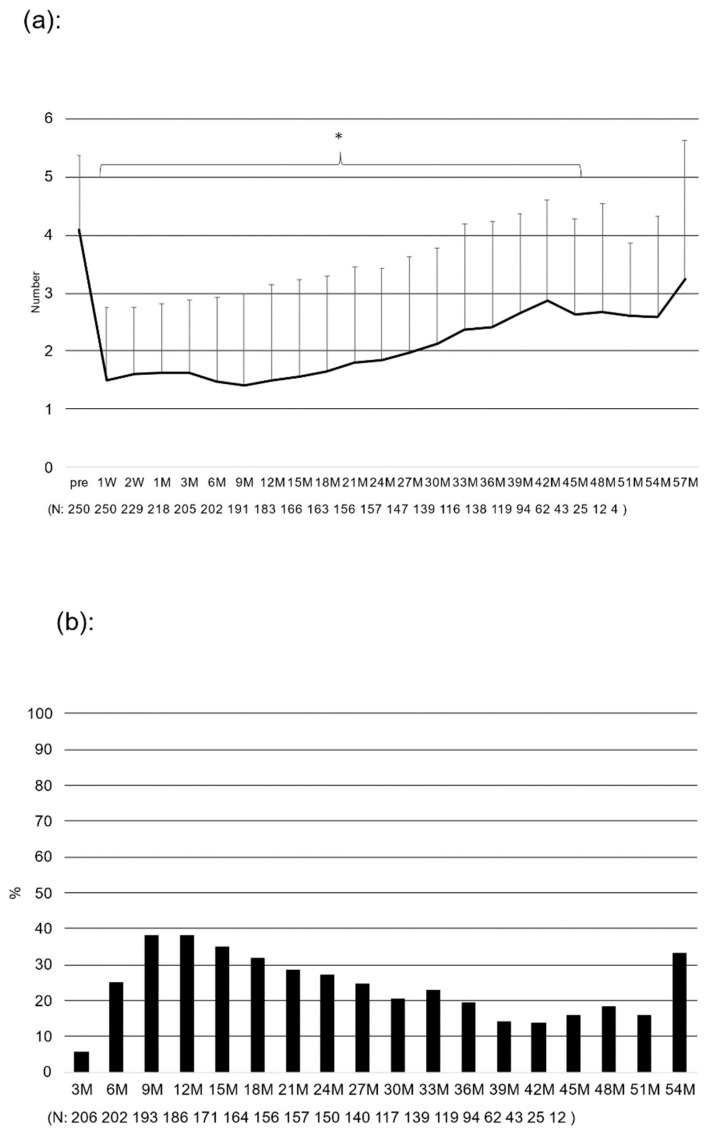
Postoperative changes in the number of antiglaucoma medications and the rate of medication-free eyes: (**a**) postoperative reduction in the number of antiglaucoma medications, * *p* < 0.001 vs. preoperative value (repeated ANOVA), (**b**) postoperative rates of medication-free eyes. Bar = SD.

**Figure 3 jcm-12-02332-f003:**
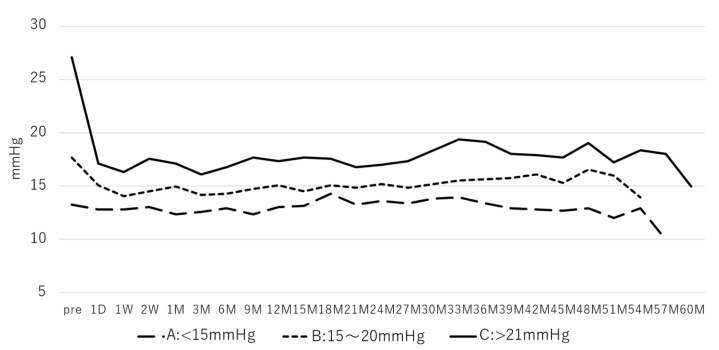
Postoperative changes: comparison by pre-IOP values: Group B: 15–20 mmHg and Group C: 21 mmHg or higher showed a significant decrease in postoperative IOP for the majority of the time periods (Group B: 1D-54 M (*p* < 0.05), Group C: 1D-57 M (*p* < 0.05)), whereas Group A: <15 mmHg showed no significant descent in all measurement periods (repeated ANOVA).

**Figure 4 jcm-12-02332-f004:**
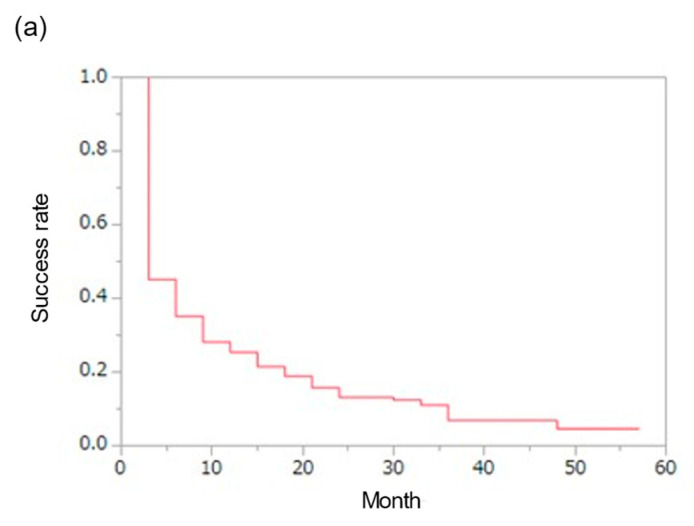
Comparison of survival curves by success definitions. (**a**) Complete success; (**b**) qualified success 1; (**c**) qualified success 2; Kaplan—Meier analysis.

**Figure 5 jcm-12-02332-f005:**
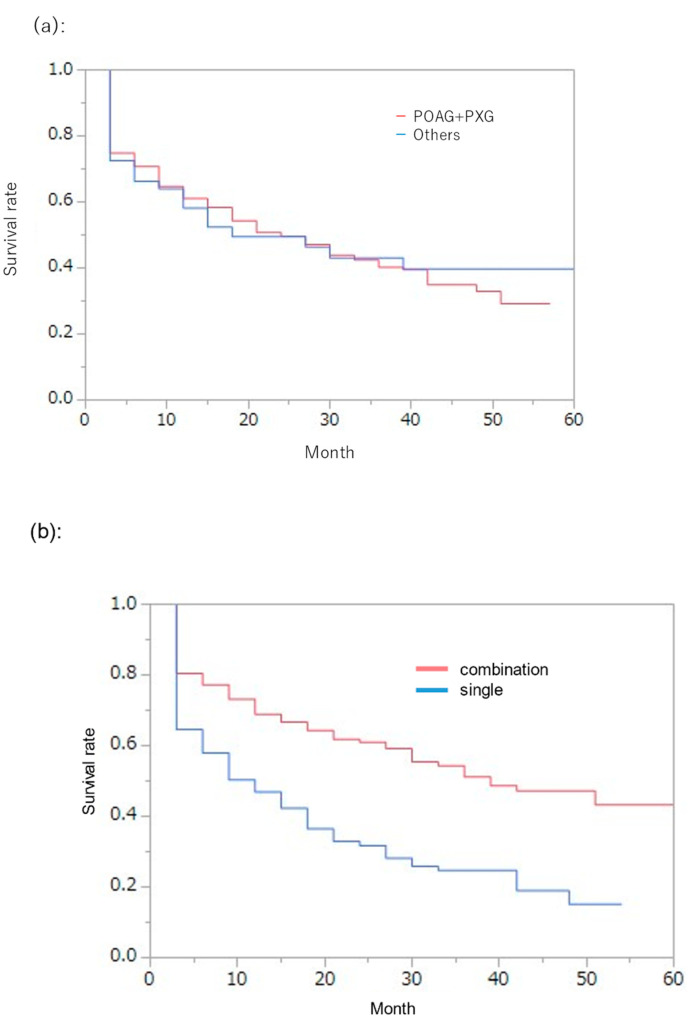
Comparison of survival curves between (**a**) type of glaucoma and (**b**) single vs. combination with cataract surgery in the qualified success 2 definition; Kaplan—Meier method.

**Figure 6 jcm-12-02332-f006:**
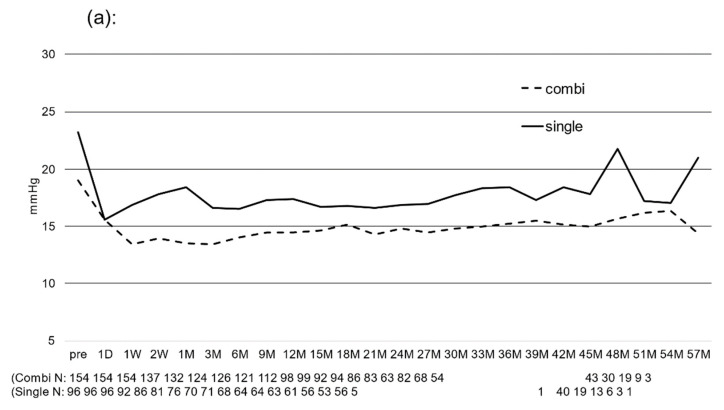
Postoperative changes in (**a**) IOP and (**b**) the number of medications between single vs. combination with cataract surgery. (**a**) IOP significantly decreased from the day after surgery to 48 months postoperatively in the combined cataract group and significantly decreased from the day after surgery to 45 months postoperatively in the trabectome^®^ alone group (*p* < 0.05). (**b**) The number of glaucoma drugs also significantly decreased from 1 week to 51 months postoperatively in the combined cataract surgery group and from 1 week to 39 months postoperatively in the trabectome^®^ alone group (*p* < 0.05).

**Figure 7 jcm-12-02332-f007:**
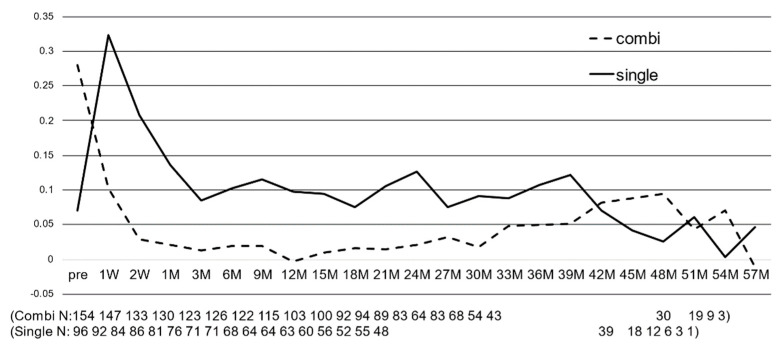
Postoperative changes in BCVA (logMAR) between single vs. combined cataract surgery. Repeated ANOVA; BCVA significantly improved from 1 to 30 months postoperatively in the combined cataract group (*p* < 0.05). In the trabectome^®^ alone group, BCVA significantly decreased during the first postoperative week (*p* < 0.001), but the significant difference from the preoperative period disappeared from 2 weeks to 57 months postoperatively.

**Table 1 jcm-12-02332-t001:** Demographics and Preoperative Ocular Parameters.

Eye (Subjects)	250 (197)	Male/Female:
		107/90
Age (range)	70.7 ± 12.6(16–93)	
Right eye:Left eye	126:124	
Glaucoma subtype	POAG	63% (158)
% (n)	PXG	16% (41)
	SOAG	12% (31)
	NVG	1% (2)
	Mixed	3% (8)
	Others	4% (10)
Pre-operative lens status	Phakia (201):IOL (48)	
Phaco combi:Single	154:96	
IOP (mmHg) (range)	20.6 ± 6.8 (11–58)	
BCVA (logMAR)	0.20 ± 0.36	
Visual field test (HFA24-2)	−10.5 ± 7.0 (1.31–−31.48)	
MD (dB) (range)		
Number of medications	4.1 ± 1.3	

POAG: primary open angle glaucoma, PXG: pseudoexfoliation glaucoma, SOAG: secondary open angle glaucoma, NVG: neovascular glaucoma, IOP: intraocular pressure, BCVA: best corrected visual acuity, LogMAR: logarithm of the minimum angle of resolution, HFA: Humphrey Field Analyzer, MD: mean deviation, numbers after ±: standard deviation.

**Table 2 jcm-12-02332-t002:** Surgical complications and postoperative procedures.

	Number	%
Endophthalmitis	0	0
Choroidal hemorrhage	0	0
IOP elevation	28	11
(IOP ≧ 30 mmHg)		
Hypotomy (<5 mmHg)	0	0
Anterior chamber irrigation	3	1
due to severe hyphema		
Additional surgery	24	10
	(trabeculectomy 19, trabectome 3, trabectome + GSL 1, trabeculotomy 1)	

GSL: goniosynechialysis.

**Table 3 jcm-12-02332-t003:** Changes of Corneal endothelial cell density.

	Pre-Operation	Post-Operation	Reduction Rate (%)	*p* Value
Total	2465.6 ± 376.3	2378.6 ± 378.2	−3.52	<0.001
Combi.	2518.4 ± 333.6	2381.0 ± 359.9	−5.45	<0.001
Single.	2353.4 ± 435.5	2374.6 ± 408.2	0.9	0.69

## Data Availability

Not applicable.

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
