# Peer review of "Mid-Term Results of Ab Interno Trabeculectomy among Japanese Glaucoma Patients"

_jcm, 2023, doi:10.3390/jcm12062332_

Round 1

Reviewer 1 Report

The authors reported the mid-term results of trabetome in japanese patients. 

1. abstract conclusion:  " safety surgery " need to change to  "safe surgery"  

    Moreover, abstract conclusion is too simple, which is different from the       conclusion in the manuscript. Please revise the conclusion in abstract.  

2. figure 2, a)Postoperative reduction of the number of anti-glaucoma 

  please add "medication" in the figure caption.

3. figure 3  pre-IOP value: need to clarify the meaning of "pre-IOP value"  which means last preoperative visit IOP or highest IOP before the surgery?

4. page 8 PE group.  Please clarify the meaning of PE group.  

 5. page 14, please check that in group C, there was no significant decrease in postoperative periods.  That is different from the results. 

6. After about 50 months in figure 1 and 2, the effect of surgery is no longer statitically significant. What is your explanation for that?

Author Response

To Reviewer 1

Thank you for your comments. I revised our paper according to your comments as indicated below.

Reviewer’s comment

  1. abstract conclusion:  " safety surgery " need to change to  "safe surgery"  

Moreover, abstract conclusion is too simple, which is different from the       conclusion in the manuscript. Please revise the conclusion in abstract.  

Author’s reply

I changed suggested part and revised our conclusion.

Reviewer’s comment

  1. figure 2, a) Postoperative reduction of the number of anti-glaucoma 

please add "medication" in the figure caption.

Author’s reply

I revised as your commented.

Reviewer’s comment

  1. figure 3  pre-IOP value: need to clarify the meaning of "pre-IOP value"  which means last preoperative visit IOP or highest IOP before the surgery?

Author’s reply

As you commented, “pre-IOP value” means last preoperative visit IOP. I revised this.

Reviewer’s comment

  1. page 8 PE group.  Please clarify the meaning of PE group.  

Author’s reply

PE group means pseudo-exfoliative glaucoma (PXG) group. I revised figure5(a).

Reviewer’s comment

  1. page 14, please check that in group C, there was no significant decrease in postoperative periods.  That is different from the results. 

Author’s reply

I revised this as suggested.

In group C, the number of glaucoma medications decreased significantly, however IOP was no significant decrease in all postoperative periods.

Reviewer’s comment

  1. After about 50 months in figure 1 and 2, the effect of surgery is no longer statitically significant. What is your explanation for that?

Author’s reply

I believe that the reason why the significant difference disappeared after 50 months was due to the decrease in the number of subjects for analysis. I added the following sentence in revised paper.

It is thought that the reason why the significant difference disappeared after 50 months was due to the decrease in the number of subjects for analysis.

Reviewer 2 Report

Mid-term results of ab interno trabeculectomy among Japanese glaucoma patients

This retrospective study assessed the effectiveness and safety of Trabectome in Japanese patients. The study had a sufficient number of participants. However, the publication included a few analytical comparisons that may have departed from the study's primary objective. Moreover, the statistical analysis may be inappropriate.

Additional comments are listed below:

1.     The manuscript was far too lengthy. Several sections might be condensed. Figure and text repeated one another. Instead of narration, the results on pages 7 to 9 should be presented using a bar chart.

2.     The study included both eyes from the same patients. The statistical analysis should account for correlated data.

3.     Corneal endothelial cell before and after surgery was analyzed using a t test. Did the author use t test or pair t test?

4.     The study aimed to report the mid-term outcomes of Trabectome. However, the authors also compared the outcomes between (1) combined phaco/Trabectome and Trabectome alone and (2) types of glaucoma. The manuscript did not have baseline characteristics for the comparing arms; thus, it is difficult to assess whether the baselines were balanced and whether the statistical methods were appropriate.

5.     What were the Trabectome indications employed at the study's institution? The study includes NVG and SOAG, which are not commonly indicated for Trabectome.

6.     If the authors intend to compare combined phaco/Trabectome vs Trabectome alone, they must explain how the patients were chosen to receive which treatment. This could be a source of bias.

7.     Twenty-four eyes underwent additional surgeries. Were these eyes included in the analysis, or were they excluded?

8.     Figure 5(a) What did PE stand for?

Author Response

To Reviewer 2

Thank you for your comments. I revised our paper according to your comments as indicated below.

Reviewer’s comment

This retrospective study assessed the effectiveness and safety of Trabectome in Japanese patients. The study had a sufficient number of participants. However, the publication included a few analytical comparisons that may have departed from the study's primary objective. Moreover, the statistical analysis may be inappropriate.

Author’s reply

Thank you for your comment, but I appreciate if your point out any specific areas for revising paper, because I consulted an expert in the field of statistics before submission.

Reviewer’s comment

The study included both eyes from the same patients. The statistical analysis should account for correlated data.

Author’s reply

Thank you for your comment. As you pointed out, I employed both eye data for the analysis, because some patients had different surgery and/or different lens status between both eyes. For example, one eye had simple surgery and the other eye had combined surgery. Or one eye was phakic eye and the other was pseudo-phakic eye. Under current condition, it is difficult to employ one-eye from one-person policy for the analysis. I appreciate your understanding.

Reviewer’s comment

Corneal endothelial cell before and after surgery was analyzed using a t test. Did the author use t test or pair t test?

Author’s reply

I used the paired t test for analysis of changes in corneal endothelial cell density.

Reviewer’s comment

The study aimed to report the mid-term outcomes of Trabectome. However, the authors also compared the outcomes between (1) combined phaco/Trabectome and Trabectome alone and (2) types of glaucoma. The manuscript did not have baseline characteristics for the comparing arms; thus, it is difficult to assess whether the baselines were balanced and whether the statistical methods were appropriate.

Author’s reply

I added to the baseline characteristics for (1) combined phaco/Trabectome and Trabectome alone and (2) types of glaucoma.

Reviewer’s comment

What were the Trabectome indications employed at the study's institution? The study includes NVG and SOAG, which are not commonly indicated for Trabectome.

Author’s reply

I perform trabectome® for open-angle glaucoma without extensive peripheral anterior synechia (PAS). In the case of NVG, Trabectome is performed if there is no active angle neovascularization.

Reviewer’s comment

If the authors intend to compare combined phaco/Trabectome vs Trabectome alone, they must explain how the patients were chosen to receive which treatment. This could be a source of bias.

Author’s reply

In the case of pseudo-phakic eyes before surgery, trabectome® alone is indicated. In the case of phakic eyes before surgery, combined cataract surgery with trabectome® is indicated when cataract is recognized. The presence or absence of cataract is determined by the surgeon before surgery. I added these points in a revised paper.

Reviewer’s comment

Twenty-four eyes underwent additional surgeries. Were these eyes included in the analysis, or were they excluded?

Author’s reply

I excluded from analysis at the time of additional surgery. I added this point in a revised paper.

Reviewer’s comment

Figure 5(a) What did PE stand for?

Author’s reply

PE group means pseudo-exfoliative glaucoma (PXG) group. I revised figure5(a).

Reviewer 3 Report

1- Why the authors involved cases with NVG? was it the open angle stage? 

2- Kindly comment on the anesthesia in the surgical technique. 

3- I felt a contradiction between these 2 statements:  

Page 15: Paragraph 3 in the study limitation: 

The authors mentioned “Trabeculectomy should be considered for patients whose IOP before surgery is stable at 15 mmHg or lower but whose visual field is progressing or whose glaucoma is at the end stage and whose target IOP of approximately 10 mmHg is desired”

Page 16:  at the conclusion: the authors mentioned that: “However, there is a limit to the amount of IOP reduction, so the indication for trabectome surgery should be carefully evaluated in patients with low target IOP” 

Author Response

To Reviewer 3

Thank you for your comments. I revised our paper according to your comments as indicated below.

Reviewer’s comment

1- Why the authors involved cases with NVG? was it the open angle stage? 

Author’s reply

We include patients with NVG showing burn-out status of their NGV and we confirmed that their angle was open without severe PAS formation.

We added the following description.

Patients with NVG showed burn-out status of their NGV and we confirmed that their angle was open without severe PAS formation.

Reviewer’s comment

2- Kindly comment on the anesthesia in the surgical technique. 

Author’s reply

We used topical oxybuprocaine hydrochloride eyedrops for anethesizing.

So, we added this information.

Reviewer’s comment

3- I felt a contradiction between these 2 statements:  

Page 15: Paragraph 3 in the study limitation: 

The authors mentioned “Trabeculectomy should be considered for patients whose IOP before surgery is stable at 15 mmHg or lower but whose visual field is progressing or whose glaucoma is at the end stage and whose target IOP of approximately 10 mmHg is desired”

Author’s reply

Thank you for your comment. We basically consider that this surgery is suitable for patients with relatively stable condition and their target intraocular pressure at mid-teen or higher. So, I revised as below/

“Trabeculectomy should be carefully considered for patients whose IOP before surgery is stable at 15 mmHg or lower but whose visual field is progressing or whose glaucoma is at the end stage and whose target IOP of approximately 10 mmHg is desired”

Page 16:  at the conclusion: the authors mentioned that: “However, there is a limit to the amount of IOP reduction, so the indication for trabectome surgery should be carefully evaluated in patients with low target IOP”